# Productivity and Community Composition of Low Biomass/High Silica Precipitation Hot Springs: A Possible Window to Earth’s Early Biosphere?

**DOI:** 10.3390/life9030064

**Published:** 2019-07-29

**Authors:** Jeff R. Havig, Trinity L. Hamilton

**Affiliations:** 1Department of Earth and Environmental Sciences, University of Minnesota, Minneapolis, MN 55455, USA; 2Department of Plant and Microbial Biology, University of Minnesota, St. Paul, MN 55108, USA; 3BioTechnology Institute, University of Minnesota, St. Paul, MN 55108, USA

**Keywords:** low biomass, hot springs, silica precipitating, carbon uptake, early Earth

## Abstract

Terrestrial hot springs have provided a niche space for microbial communities throughout much of Earth’s history, and evidence for hydrothermal deposits on the Martian surface suggest this could have also been the case for the red planet. Prior to the evolution of photosynthesis, life in hot springs on early Earth would have been supported though chemoautotrophy. Today, hot spring geochemical and physical parameters can preclude the occurrence of oxygenic phototrophs, providing an opportunity to characterize the geochemical and microbial components. In the absence of the photo-oxidation of water, chemoautotrophy in these hot springs (and throughout Earth’s history) relies on the delivery of exogenous electron acceptors and donors such as H_2_, H_2_S, and Fe^2+^. Thus, systems fueled by chemoautotrophy are likely energy substrate-limited and support low biomass communities compared to those where oxygenic phototrophs are prevalent. Low biomass silica-precipitating systems have implications for preservation, especially over geologic time. Here, we examine and compare the productivity and composition of low biomass chemoautotrophic versus photoautotrophic communities in silica-saturated hot springs. Our results indicate low biomass chemoautotrophic microbial communities in Yellowstone National Park are supported primarily by sulfur redox reactions and, while similar in total biomass, show higher diversity in anoxygenic phototrophic communities compared to chemoautotrophs. Our data suggest productivity in Archean terrestrial hot springs may be directly linked to redox substrate availability, and there may be high potential for geochemical and physical biosignature preservation from these communities.

## 1. Introduction

Terrestrial and submarine hot springs may have been ideal locations for early life on Earth. Today, microbial life thrives in geochemically and geographically distinct hydrothermal springs in terrestrial and submarine settings. These observations coupled to terrestrial hot springs dating to 3.48 Ga [1,2] provide ample evidence for niche space on land masses early in the evolution of life. Evidence for hot spring deposits have been found on Mars including in the Gusev Crater and other sites [3,4,5,6,7,8,9]. The terrestrial hot spring deposits reported by [1,2] likely pre-date the evolution of oxygenic photosynthesis (evidence for oxidative weathering does not appear in the rock record until ~3.0 Ga [10,11]) and may have formed close to the initial evolution of anoxygenic photosynthesis (e.g., [12,13]). Thus, microbial life inhabiting terrestrial hot springs early in the Archean was likely supported by chemolithoautotrophy or anoxygenic photosynthesis.

Microbial communities present in hot spring environments on the early Earth would have been substrate limited, depending upon the delivery of exogenous electron acceptors and donors to generate energy and drive autotrophy to fix CO_2_ into organic carbon to create biomass. This substrate (and thus energy) limitation would have likely resulted in low-biomass microbial communities compared to microbial communities based on oxygenic photosynthesis, which are nutrient (e.g., N, P, Fe) limited rather than energy limited and can generate much larger quantities of fixed carbon/biomass. With a need to better understand the types of microbial communities that may best represent what would have been prevalent in hot springs on the early Earth and potentially Mars, we need to characterize systems that can serve as potential analogs: microbial communities rooted in chemolithoautotrophy or anoxygenic photosynthesis.

Recent hypotheses for a hot spring origin of life (e.g., [14,15,16,17,18,19,20]) also provide impetus for characterization of low biomass geothermal springs fueled by chemolithoautotrophy. In addition, the Mars 2020 rover mission has been charged with searching for evidence of past life on the Martian surface at Jezero Crater, where a long history of impact and volcanism-generated heat could have driven hydrothermal processes (e.g., [21,22,23]). Extant hot springs provide analog systems for both early Earth hot spring microbiota and the potential for Martian geothermal life. Studies of hot springs on Earth have the potential to inform the Mars2020 mission should the rover encounter hydrothermal deposits.

Earliest Archean hot springs were likely dominated by chemoautotrophs, which in modern hot springs support relatively low biomass, particularly in sites not supported by photoautotrophy [24,25]. Characterization of the taxonomy and physiology of extant low biomass hydrothermal systems may aid in interpreting early life on earth, how this life was preserved in the silica depositing systems, and inform what we might expect to find in putative hot spring deposits on Mars. It is our hypothesis that for chemolithoautotrophic and anoxygenic photoautotrophy, chemical substrates required as electron donors for the reduction of CO_2_ (e.g., H_2_, S^2−^) are a fundamental limiting factor for productivity.

In this study, we sampled low biomass (≤ 1% organic C by dry weight) hot springs across a range of pH values (pH 3.0 to 8.6) at temperatures above the upper limit for photosynthesis (> 72 °C) as well as select low biomass sites with photoautotrophy for comparison. For all sites, we characterized the geochemistry and community composition, and measured carbon uptake rates (rates of autotrophy). Our data suggests low biomass hydrothermal microbial communities are substrate-limited (e.g., chemolithoautotrophs and anoxygenic photoautotrophs), which limits productivity. We report evidence for distinct co-occurring chemolithoautotrophic microbial consortia that drive both sulfur oxidizing and sulfur reducing reactions, providing evidence for sulfur cycling at the microscale. Finally, comparison of a low biomass oxygenic photoautotrophic microbial community to putative anoxygenic photoautotrophic communities suggest that while carbon uptake rates of oxygenic photosynthesis is over an order of magnitude greater, putative anoxygenic phototrophic communities are more diverse than both oxygenic phototrophic and chemolithoautotrophic microbial communities. All of these results provide a starting point for understanding and characterizing the types of environments where Earth’s earliest biosphere may have thrived.

## 2. Methods

### 2.1. Geochemistry

Water samples were collected and analyzed as described previously [26,27,28]. In short, temperature and pH were measured onsite using a WTW 330i meter and probe (Xylem Analytics, Weilheim, Germany) and conductivity was measured with a YSI 30 conductivity meter and probe (YSI Inc., Yellow Springs, OH). Sulfide, ferrous iron (Fe^2+^) and Silica were measured onsite using a DR1900 portable spectrophotometer (Hach Company, Loveland, CO). Water samples were filtered through 25 mm diameter 0.2 µm polyethersulfone syringe filters (VWR International, Radnor, PA) using a 140 mL syringe that had been acid washed (three-day soak in 10% TraceMetal Grade HNO_3_ (Fisher Scientific, Hampton, NH) followed by triple rinsing with 18.2 MΩ/cm deionized (DI) water). Anion samples for ion chromatography (IC) were filtered into filtered into 15-mL centrifuge tubes. Samples for induced coupled plasma optical emission spectroscopy (ICP-OES) and induced coupled plasma mass spectrometry (ICP-MS) were filtered into a 15 mL centrifuge tubes that had been acid washed, and acidified (400 µL of concentrated OmniTrace Ultra™ concentrated nitric acid (EMD Millipore, Billerica, MA)). Samples collected in 2016 for anion and cation/trace element analyses were analyzed as previously described [26]. All samples collected in 2017 and 2018 for IC and ICP analyses were kept on ice/refrigerated at 4 °C in the dark until analysis. IC analyses for Cl^−^ and SO_4_^2−^ was conducted via a Thermo Scientific Dionex ICS 5000+ ion chromatography system; ICP-OES analysis for Na, K, Ca, Mg, and P was carried out using a Thermo Scientific iCAP 6000 series ICP-OES; and ICP-MS analysis for Al, Mn, Fe, and As was conducted using a standard addition method on a Thermo Scientific X Series 2 ICP-MS, all at the Analytical Geochemistry Laboratory in the Department of Earth Sciences at the University of Minnesota.

Samples for dissolved inorganic carbon (DIC) analysis were filtered into a gas-tight syringe and then injected into Labco Exetainers^®^ (Labco Limited, Lampeter, UK) pre-flushed with He, with excess He removed following introduction of 4 mL of filtered sample with minimal agitation. Samples were stored inverted on ice/refrigerated at 4 °C until returned to the lab, where 1 mL of concentrated H_3_PO_4_ was added and the samples shipped to the Stable Isotope Facility (SIF) at the University of California, Davis for analysis. DIC analysis for concentration and ^13^C isotopic signal using a GasBench II system interfaced to a Delta V Plus isotope ratio mass spectrometer (IR-MS) (Thermo Scientific, Bremen, Germany) with raw delta values converted to final using laboratory standards (lithium carbonate, δ^13^C = −46.6‰ and a deep seawater, δ^13^C = +0.8‰) calibrated against standards NBS-19 and L-SVEC. Samples for dissolved organic carbon (DOC) analysis were filtered through a 0.2 µm polyethersulfone syringe filter that had been flushed with ~30 mL of sample and then ~40 mL put into a 50 mL centrifuge tube and then immediately flash-frozen on dry ice and kept frozen and in the dark until analysis at the SIF. DOC analysis for concentration and ^13^C isotopic signal were carried out using O.I. Analytical Model 1030 TOC Analyzer (O.I. Analytical, College Station, TX) interfaced to a PDZ Europa 20-20 isotope ratio mass spectrometer (Sercon Ltd., Cheshire, UK) utilizing a GD-100 Gas Trap Interface (Graden Instruments) for concentration and isotope ratio determination with raw delta values converted to final using laboratory standards (KHP and cane sucrose) calibrated against USGS-40, USGS-41, and IAEA-600.

### 2.2. Imaging and Elemental Quantification via SEM-EDX

Imaging and elemental quantification of sediment and biofilm samples was carried out via scanning electron microscopy (SEM) with energy dispersive X-ray spectroscopy (EDX). The SEM used was a Hitachi TM-1000 (Hitachi High-Technologies Corporation, Japan), with an attached SwiftED-TM EDX (Oxford Instruments, UK), owned and maintained by the National Lacustrine Core Facility (LacCore) within the Department of Earth and Environmental Sciences at the University of Minnesota. The SEM beam has a fixed 15 kV acceleration energy, with a pre-centered tungsten hairpin type electron gun. The specimen chamber vacuum is held between 30 and 50 Pa. Samples for analysis via SEM-EDX were flash frozen on dry ice in the field, and were kept frozen until analysis. Wet samples were placed onto 1-inch diameter round glass slides and dried. No coating was applied to the samples prior to analyses. General classification of features observed in SEM images (with associated EDX spectra shown) is discussed in detail in the SOM. Each sample was observed using SEM for a minimum of one and a half hours, conducting imaging and EDX spot analyses across the entirety of each sample to characterize co-occurrences and elemental compositions of features observed.

### 2.3. Photoassimilation of CO_2_ and Analysis of Solid Samples for ^13^C and ^15^N

Inorganic carbon uptake was assessed in situ using a microcosm-based approach through the addition of NaH^13^CO_3_. Samples were collected using pre-sterilized spatula or forceps and ~500-mg was placed into pre-combusted (12 h, 450 °C) serum vials or sterile 6-well flat bottom cell culture plates (Fisher Scientific, Pittsburgh, PA). Samples were overlaid with spring water from the collected site and serum vials were capped with gas-tight black butyl rubber septa. Assays were initiated by addition of NaH^13^CO_3_ (Cambridge Isotope Laboratories, Inc., Andover MA) (100 µM final concentration). Each sample was placed into one six well tray per site. At each site, we assessed the potential for photoautotrophic (light) and chemoautotrophic (dark) NaH^13^CO_3_ uptake. To assess CO_2_ assimilation in the dark, vials (n = 3 replicates) were amended with NaH^13^CO_3_ and completely wrapped in aluminum foil. All reported values of DIC uptake (carbon fixation rates) reflect the difference in uptake between the labeled and unlabeled assays. Following incubation time (t ≈ 2 h), incubation trays were flash frozen on dry ice and kept frozen until processed.

Following return to the lab, incubation samples were rinsed with 1 M HCl to remove any carbonate minerals and any residual ^13^C DIC label, triple rinsed with 18.2 MΩ/cm DI water and dried (8 h, 60 °C). Natural abundance C and N contextual samples were collected in 5 mL cryovials, flash-frozen on dry ice, and kept frozen until processed for analysis. Natural abundance samples were dried (8 h, 60 °C). Natural abundance samples and incubation samples were ground and homogenized using a clean mortar and pestle (silica sand and 80% ethanol slurry ground between samples, then mortar and pestle rinsed with ethanol followed by 18.2 MΩ/cm DI water and dried). For samples collected in 2016, analyses were conducted as previously reported [26]. For samples collected in 2017 and 2018, solid samples are analyzed using an Elementar Vario EL Cube or Micro Cube elemental analyzer (Elementar Analysensysteme GmbH, Hanau, Germany) interfaced to a PDZ Europa 20-20 isotope ratio mass spectrometer (Sercon Ltd., Cheshire, UK) by the Stable Isotope Facility at the University of California, Davis. Samples are combusted at 1080 °C in a reactor packed with copper oxide and tungsten (VI) oxide. Following combustion, oxides were removed in a reduction reactor (reduced copper at 650 °C). The helium carrier then flows through a water trap (magnesium perchlorate). N_2_ and CO_2_ are separated using a molecular sieve adsorption trap before entering the IRMS. During analysis, samples are interspersed with several replicates of at least two different laboratory standards (bovine liver, glutamic acid, enriched alanine, nylon 6) which have been previously calibrated against NIST Standard Reference Materials (IAEA-600, USGS-40, USGS-41, USGS-42, USGS-43, USGS-61, USGS-64, and USGS-65).

Total uptake of DIC was calculated using DIC numbers determined for the source water (described above). All stable isotope results are given in delta formation expressed as per mil (‰). Carbon stable isotopes are calculated as:δ^13^C = [((R_a_)_sample_ / (R_a_)_standard_) − 1] × 10^3^,(1)
where R_a_ is the ^13^C / ^12^C ratio of the sample or standard, and are reported versus the Vienna Pee Dee Belemnite (VPDB) standard. Natural abundance samples also had δ^15^N values determined as described for δ^13^C, reported versus atmospheric air.

### 2.4. DNA Extraction and Quantification

DNA was extracted in triplicate from ~250 mg of biomass using a DNeasy PowerSoil Kit (Qiagen, Carlsbad, CA) according to the manufacturer’s instructions. Equal volumes of each extraction were pooled and the concentration of DNA was determined using a Qubit dsDNA HS Assay kit and a Qubit 3.0 Fluorometer (Invitrogen, Burlington, ON, Canada). As a negative DNA extraction control, DNA was extracted from the filter used for the field blank water sample (described above). No DNA was detected in the control and sequencing failed to generate amplicons (see below for amplicon sequencing details).

### 2.5. Sequence Analysis

Bacterial and archaeal 16S rRNA gene sequences were targeted using the primers 515f and 806rB [29,30]. For sequencing, total DNA was submitted to the University of Minnesota Genomics Center and amplicons were sequenced using MiSeq Illumina 2 × 300 bp chemistry. Each sample was sequenced once. Post sequence processing was performed within the Mothur (ver. 1.39.3) sequence analysis platform [31] following the MiSeq SOP [32]. Read pairs were assembled and contigs were trimmed to include only the overlapping regions and unique sequences were aligned against a SILVA-based reference alignment and classified using a Bayesian classifier within Mothur against the against the Silva (v132) reference taxonomy. Chimeras were identified and removed using UCHIME [33]. Operational Taxonomic Units (OTUs) were assigned to all classified sequences at a sequence similarity of 97.0% for archaea and bacteria. Rarefaction was calculated within Mothur and based on rarefaction analysis, > 95% of the predicted 16S gene diversity was sampled at this depth of sequencing (data not shown). Alpha diversity measurements samples were calculated from rarefied using the R package Phyloseq (ver. 1.16.2; [34]). Sequence data including raw reads, quality scores and mapping data have been deposited in the NCBI Sequence Read Archive (SRA) database with the BioProject number PRJNA395419.

## 3. Results

### 3.1. Site Descriptions

Six sites across Yellowstone National Park were selected for sampling based on criteria of temperature (above and below the upper temperature limit for photosynthesis, or 72 °C), pH (acidic to circum-neutral to alkaline), and low biomass (measured as organic carbon ≤ 1% C by dry weight). Using these parameters as a guide, we selected three sites in the Sylvan Springs Area (SSA) of the Gibbon Geyser Basin (GGB), two sites in the Rabbit Creek Area (RCA) of the Midway Geyser Basin (MGB), and one site in the Imperial Geyser Basin (IGB) of the Lower Geyser Basin (LGB), all in Yellowstone National Park (YNP), WY, USA (Figure 1A).

### 3.2. Rabbit Creek Area, MGB

‘Heartbeat Pool’ (no official name or designation) was a round pool with a diameter of approximately 2 m (Figure 1B). The colloquial name was given to it based on a periodic deep thumping sound (presumed driven by subsurface boiling) that cause the surface to pulse. Dispersed diffuse gas release occurred, and no discernable outflow was observed. The hot spring pool had a temperature of 79.2 °C and a pH of 3.01 at the time of sampling. There were no visually apparent microbial features along the edge of the hot spring (e.g., filaments or mats), and loose siliceous mud was collected for sampling and imaging (Figure 1C, arrow). ‘Rose Terrace Pool’ (no official name or designation) was a small (approximately 0.5 m across at the longest point) roughly square-shaped pool with an off-centered source and inflow from a smaller pool with an assumed separate source, a small outflow channel, and was ringed with white scallop-shaped silica precipitate along the edge of the pool and channel (Figure 1D). The hot spring pool had a temperature of 74.8 °C and a pH of 8.62 at the time of sampling. There were no visually apparent microbial features along the edge of the hot spring (e.g., filaments or mats), and loose siliceous mud was collected for sampling and imaging (Figure 1D, arrow).

### 3.3. Imperial Geyser Basin, LGB

Boulder Geyser (official YNP Research Coordination Network designation: LRNN781) has a continuously geysing source that is at or above the boiling point of water (≥ 93.3 °C), and feeds a large outflow channel (~2.4 m wide) that has black, loosely-consolidated sediments in the chemotrophic zone (Figure 1E). The hot spring outflow had a temperature of 84.9 °C and a pH of 8.47 at the time and location of sampling, which was approximately 9.4 m downstream from the source. There were no visually apparent microbial features observed associated with the black sediment (e.g., filaments or mats), and loose siliceous mud was collected for sampling and imaging (Figure 1E inset, arrow).

### 3.4. Sylvan Spring Area, GGB

‘The Dryer’ (official YNP Research Coordination Network designation: GSSGNN037) was an approximately 1 m by 3 m hot spring with a continuous low flow outflow with no discernable gaseous discharge (e.g., gas bubbles) and an outflow armored with silica precipitate (Figure 1F). The hot spring pool had a temperature of 40.5 °C and a pH of 7.13 at the time of sampling. There was a visually detectable greenish tint to the loose sediments at the water-sediment interface, and this loose silica precipitate was collected for sampling and imaging (Figure 1F, insert, arrow). ‘The Dryer’ had a thick photosynthetic mat that developed in 2010–2011, but was not present previously and was gone in 2012, leaving behind a thick silica precipitate preserving the mat textures (Appendix A). ‘Avocado Spring’ (official YNP Research Coordination Network designation: GSSGNN025) was an approximately 5 m by 7 m hot spring with a continuously roiling volcanic-gas-driven zone at the side farthest from the outflow channel, driving mixing of the hot spring water, and a low flow outflow channel with a loosely-consolidated microbial mat that transitioned from dark grey closer to the source to reddish-grey further down the outflow channel (Figure 1G). The hot spring outflow channel had a temperature of 69.5 °C and a pH of 6.39 at the ‘grey mat’ location sampled in 2016 and a temperature of 65.6 °C and a pH of 6.71 at the ‘red mat’ location sampled in 2018 (Figure 1G, insert, arrows). Dante’s Inferno (official YNP name and Research Coordination Network designation) was an approximately 12 m by 17 m hot spring with a continuously roiling volcanic-gas-driven zone in the center of the hot spring driving mixing of the hot spring water, and a terraced outflow of silica precipitate (Figure 1H). The hot spring pool had a temperature of 79.4 °C and a pH of 5.19 at the time of sampling. There were no visually apparent microbial features along the edge of the hot spring (e.g., filaments or mats), and silica precipitate from the surface of the armored pool edge was collected for sampling and imaging (Figure 1I, arrow). 

### 3.5. Geochemistry

All physical and aqueous geochemistry results are reported in Table 1. Temperature of the sites range between 40.5 °C and 84.9 °C and pH values between 3.01 and 8.62. Conductivity ranged from 505 to 3596 µS/cm. Dissolved silica (SiO_2(aq)_) concentration were all high, with values from 2.19 to 4.83 mmol/L. Chloride concentration was lowest at ‘Heartbeat Pool’, with a value of 0.24 mmol/L, and highest at ‘The Dryer’, with a value of 15.95 mmol/L. Sulfate concentration was lowest at ‘Rose Terrace Pool’ (0.14 mmol/L) and highest at ‘Avocado Spring’ (3.27 mmol/L). Sodium and potassium were lowest at ‘Heartbeat Pool’ (0.81 and 0.17 mmol/L, respectively), while magnesium and calcium were lowest at ‘Rose Terrace Pool’ (0.21 and 14.31 µmol/L, respectively). The highest concentrations for these cations were found at ‘The Dryer’ (17.61 mmol/L Na, 128.98 µmol/L Ca), Dante’s Inferno (1.53 mmol/L K), and ‘Heartbeat Pool’ (7.90 µmol/L Mg). Sulfide concentration ranked from 1.0 µmol/L at ‘Rose Terrace Pool’ to 538.0 µmol/L at Dante’s Inferno. Ferrous iron (Fe^2+^) was below detection limits at most sites except at ‘Avocado Spring’ and Boulder Geyser outflow where concentrations of 0.10 and 0.90 µmol/L were observed. Phosphorous was not determined at all sites, but when analyses were done, concentrations ranged from a low of 2.40 µmol/L at ‘Heartbeat Pool’ to a high of 8.96 µmol/L at ‘Avocado Spring’. Aluminum concentration was highest at ‘Heartbeat Pool’ (42.01 µmol/L), and lowest at ‘The Dryer’ (3.38 µmol/L). Manganese concentration ranged from a low of 7.71 nmol/L at ‘Rose Terrace Pool’ to a high of 684 nmol/L at Dante’s Inferno, while total iron concentration was highest at ‘Heartbeat Pool’ (507 nmol/L) and lowest at ‘Avocado Spring’ (85 nmol/L). Arsenic concentrations were consistently high (≥ 14,000 nmol/L) for most sites, with the highest at 34,269 nmol/L at ‘The Dryer’, and one low concentration site being ‘Heartbeat Pool’ at 137 nmol/L. The values for most of the hot springs (‘Heartbeat Pool’, Dante’s Inferno, ‘Avocado Spring’, and ‘The Dryer’) are consistent with geochemistry driven by subsurface boiling and subsequent phase separation (liquid vs. vapor phases) and differential remixing of those phases or minimal subsurface boiling and resulting phase separation (Boulder Geyser, ‘Rose Terrace Pool’) [28,35,36,37,38,39,40].

### 3.6. Inorganic Carbon Uptake and Carbon and Nitrogen Stable Isotopes

All inorganic carbon uptake results and dissolved inorganic carbon, dissolved organic carbon, and biomass results are reported in Table 1, Table 2 and Figure 2. Dissolved inorganic carbon concentration ranged from a low of 0.18 mmol/L at ‘The Dryer’ to a high of 39.11 mmol/L at ‘Rose Terrace Pool’, and dissolved organic carbon (when determined) ranged from a low of 31.9 µmol/L at Bolder Geyser outflow to a high of 195 µmol/L at Dante’s Inferno. All dissolved inorganic carbon δ^13^C values fell between −1.65 and 3.06‰ except ‘Heartbeat Pool’ where the δ^13^C of DIC was −8.44‰ (semi-quantitative due to low signal). Dissolved organic carbon δ^13^C values (when measured) ranged from −27.80 to −22.16‰, consistent with input of allochthonous organic material [25]. 

Biomass δ^13^C values fell into two groups: more positive δ^13^C samples from Dante’s Inferno (−14.69‰), ‘Avocado Spring’ red mat (−15.83‰), and ‘The Dryer’ (−16.67‰) and more negative δ^13^C samples from ‘Heartbeat Pool’ (−25.59‰), Boulder Geyser outflow (−24.33‰), ‘Rose Terrace Pool’ (−24.33‰), and ‘Avocado Spring’ grey mat (−22.87‰). Biomass δ^15^N values ranged from a low of −4.13‰ at ‘The Dryer’ to a high of 4.08‰ at ‘Avocado Spring’ red mat, with the other sites falling close to the range expected for nitrogen fixation (~−2 to 2‰). More negative δ^15^N values are consistent with uptake of fixed nitrogen from a vapor phase input source, while more positive values are consistent with an allochthonous input of fixed nitrogen and/or in situ nitrogen cycling and denitrification [25,41,42]. 

Light-dependent inorganic carbon uptake was observed at ‘Avocado Spring’ (16.22 and 22.56 µg C uptake/g C_biomass_/hr) and ‘The Dryer’ (355.35 µg C uptake/g C_biomass_/hr) while no light-independent (dark) assimilation was observed at these sites. In contrast, rates of light-dependent and light-independent inorganic carbon assimilation were indistinguishable at ‘Heartbeat Pool’, Dante’s Inferno, Boulder Geyser outflow, and ‘Rose Terrace Pool’, with average values ranging from 4.49 µg C uptake/g C_biomass_/hr at ‘Heartbeat Pool’ to 21.32 µg C uptake/g C_biomass_/hr at Dante’s Inferno. Light-dependent carbon uptake is consistent with a predominance of a photoautotrophy, while a lack of light-dependent carbon uptake is consistent with chemolithoautotrophy, and a large range of values for any single treatment was likely due to inherent heterogeneity within the sites sampled.

### 3.7. Microbial Community Composition

16S rRNA amplicon data are reported in Figure 3. Archaeal and bacterial 16S rRNA sequences were recovered from all sites sampled. Archaeal 16S rRNA recovery ranged from a high of 65,163 sequences at ‘Heartbeat Pool’ to 20 sequences at ‘The Dryer’. Bacterial 16S rRNA recovery ranged from a low of 15,247 sequences at ‘The Dryer’ to a high of 209,630 sequences at ‘Avocado Spring’ red mat. Most sites had ≥ 95% of Archaeal sequences recovered from OTUs associated with Crenarchaeota, with the two ‘Avocado Spring’ sites deviating from this. Of the non-Crenarchaeotal Phyla recovered, OTUs most closely affiliated with Diapherotrites, Euryarchaeota, Thaumarchaeota, and unclassified Archaea were predominant. A greater diversity of Bacterial phyla was recovered, with OTUs recovered most closely affiliated with Acidobacteria, Actinobacteria, Aquificae, Armatimonadetes, Bacteriodetes, Chloroflexi, Cyanobateria, Deinococcus-Thermus, Patescibacteria, Planctomycetes, Proteobacteria, Thermotogae as well as unclassified Bacteria. 

## 4. Discussion

The following section is broken down by pH (acidic, alkaline, and circum-neutral), and compares sites with microbial communities rooted in chemolithoautotrophic productivity to those with light-dependent autotrophy. 

### 4.1. Acidic Site (‘Heartbeat Pool’)—Low Biomass, Low pH

The lowest rate of inorganic carbon uptake in this study, with 4.5 to 6.3 µg C uptake/g C biomass/hr, was observed in sediments from ‘Heartbeat Pool’. Rates of light-dependent carbon assimilation were indistinguishable from microcosms performed in the dark suggesting chemoautotrophy was active in the sediments (Figure 2, Appendix A). In ‘Heartbeat Pool’, more than a third of the archaeal sequences recovered (37.1%) were assigned to Sulfolobales, including *Stygiolobus* sp. (the primary OTU) as well as *Metallosphaera* sp., *Sulfolobus* sp., and uncultured representatives (Figure 3). Most members of *Sulfolobus* are capable of chemolithoautotrophy coupled to sulfur, including oxidation of sulfide, elemental sulfur (S^0^), pyrite (FeS_2_), as well as H_2_ and Fe^2+^, using O_2_ as the electron acceptor. Most Sulfolobales are also capable of chemoheterotrophy linked to reduced sulfur species while some are strict heterotrophs [43,44,45,46]. *Stygiolobus* can also reduce S^0^ to sulfide using H_2_ as an electron donor [44]. 

The single most abundant Bacterial order recovered from ‘Heartbeat Pool’ was Acetobacteriales (22.3%) and the most abundant OTU was assigned to *Acidiphilium* (Figure 3). *Acidiphilium* sp. are capable of chemolithoautotrophy coupled to S^0^ reduction, or Fe^3+^ reduction coupled to H_2_ or organic carbon oxidation, as well as expression of bacterial chlorophyll with Zn replacing Mg in the reaction center [47,48]. Given the lack of light-dependent carbon uptake and a temperature of 79.2 °C, it is our assumption that photoautotrophy was not occurring. Thus, the recovery of large numbers of sequences associated with Chloroflexi (18.9% of recovered Bacterial sequences) are presumed to be associated with heterotrophs. For instance, 41.3% of Chloroflexi sequences were assigned to Ktedonobacteriales genus JG30-KF-AS9, an uncultured aerobic heterotroph found in mildly acidic acid rock drainage sites [49]. 

The presence of small Fe-S crystals in the sediments associated with biomass (Figure 4) suggest that generation of sulfide and Fe^2+^ was occurring, consistent with *Acidiphilium* sp. reducing Fe^III^ to Fe^2+^ and Sulfolobus (e.g., *Stygiolobus* sp.) and/or Thermoproteales (e.g., *Vulcanisaeta* sp. and *Thermoproteus* sp.) reducing S^0^ to sulfide [44,50,51]. Many sequences were recovered associated with strictly heterotrophic lineages, including bacteria (e.g., Frankiales *Acidothermus* sp., 4.9% of bacterial sequences recovered; IMCC26256, 9.0% of sequences recovered; Isophaerales *Aquisphaera* sp. and *Singulisphaera* sp., 14.9% of sequences recovered; *Ktedonobacterales* JG30-KF-AS9 sp., JG30a-KF-32 sp., *Thermogemmatispora* sp., 16.8% sequences recovered; and Solirubrobacterales *Conexibacter* sp., 8.7% of sequences recovered) and Archaea (Geoarchaeales sp., 41.3% of Archaeal sequences recovered). The recovery of large numbers of sequences affiliated with known or putative heterotrophs suggest breakdown of autochthonous and/or allochthonous organic material is an important component of the microbial community at ‘Heartbeat Pool’.

### 4.2. Alkaline Sites (Boulder Geyser and ‘Rose Terrace Pool’)

Boulder Geyser and ‘Rose Terrace Pool’ had temperatures higher than the upper limit for photosynthesis, 84.8 °C and 74.8 °C, respectively (Table 1) and no light-dependent assimilation of ^13^C was observed (Figure 2, Appendix A). 

Boulder Geyser outflow sediments yielded average carbon uptake rates of 18.4 to 20.5 µg C uptake/g C biomass/hr, suggesting active chemolithoautotrophy. SEM images of the sediments revealed the prevalence of EPS in two populations, either intimately associated with Fe-S minerals (e.g., Figure 5C,D,F), or entirely exclusive of Fe-S minerals (e.g., Figure 5B and the large particle surface in Figure 5E). From these observations, we hypothesize two distinct chemolithoautotrophic populations based on the abundance of sequences *Thermocrinis* sp. (50.0% of bacterial sequences recovered), Thermosulfobacteriales (*Geothermobacterium* sp., 35.3% of bacterial sequences recovered), Desulfurococcales (*Thermosphaera* sp., *Sulfophobococcus* sp., *Ignisphaera* sp.), and Thermoproteales (*Thermofilum* sp. and *Pyrobaculum* sp., 7.8% of archaeal sequences recovered). Characterized *Thermocrinis* sp. oxidize reduced sulfur species (e.g., S^0^, S_2_O_3_^2−^) [52] which is inconsistent with Fe-S precipitation. Thus, we suggest the Fe-S devoid biomass may represent *Thermocrinis*-predominant biomass. Biomass associated with Fe-S precipitation (either diffuse or as framboids, Figure 5) would then be associated with a microbial consortium that includes Fe^III^-reducing (e.g., Geothermobacterium, [53]) as well as polysulfide, S^0^, S_2_O_3_^2−^, and SO_3_^2−^ reducing Desulfurococcales and/or Thermoproteales OTUs [54,55,56,57,58]. Furthermore, some Thermoproteales and Desulfurococcales break down organic material fermentatively, potentially generating H_2_ for Geothermobacterium as well as Thermocrinis [52,53,56,58,59]. The co-occurrence of active sulfur oxidation and sulfur reduction suggests a complex sulfur cycle is present in these types of hydrothermal systems, with the potential to influence sulfur isotope signals in the water, biomass, and minerals present. Further research is needed to test this hypothesis.

Inorganic carbon assimilation was also observed in ‘Rose Terrace Pool’ sediments albeit at lower rates than Boulder Geyser sediments (Figure 2). Sequences affiliated with *Thermocrinis* sp. were abundant (16.8% of bacterial sequences recovered) and we did not observe Fe-S minerals in ‘Rose Terrace Pool’ sediments. The presence of iron oxide minerals (Figure 6) indicated potential for Fe^2+^ oxidation, though no sequences of explicit Fe^2+^ oxidizers were recovered. The rest of the sequences recovered were primarily related to characterized heterotrophs, including Thermales (*Thermus* sp., 30.9% of Bacterial sequences recovered), Fervidibacteria (9.1% of Bacterial sequences recovered), Aigarchaeales (*Caldiarchaeum* sp., 68.8% of Archaeal sequences recovered), and Desulfurococcales (*Ignisphaera* sp., *Thermosphaera* sp., and unclassified sp., accounting for 22.9% of Archaeal sequences recovered). It should be noted, however, that a significant portion of the sequences recovered were unclassified, including over 19.2% of all Bacterial sequences and most of the Desulfurococcales Archaeal sequences recovered.

### 4.3. Circum-Neutral Sites (Dante’s Inferno, ‘Avocado Spring’, ‘The Dryer’)

We sampled circum-neutral pH sites (pH ~5 to 7) that included no visible phototrophs (Dante’s Inferno) and visible phototrophs (‘Avocado Spring’ and ‘The Dryer’) to compare chemoautotrophic versus photoautotrophic inorganic carbon uptake in low biomass sites. 

### 4.4. Chemoautotrophy

We observed inorganic carbon uptake in sediments from Dante’s Inferno (pH, 79.4 °C) suggesting an active chemolithoautotrophic microbial community is present (19.7 to 21.3 µg C uptake/g C biomass/hr, Figure 2, Appendix A). SEM imaging of the sediments revealed putative physical evidence for two microbially-mediated processes: oxidation of S^0^ spheres (S^0^ spheres coated with EPS/Biomass, smaller spheres irregularly shaped due to localized oxidation, Figure 7C,D) and precipitation of Fe-S minerals, with S^0^ oxidation likely predominant (Figure 7). It is our assumption that all autotrophic metabolisms in Dante’s Inferno are based in chemolithotrophy and are thus limited by substrate input into the system. The 16S rRNA amplicon data is consistent with our interpretation of microbial Fe and S processing at Boulder Geyser: that there are putative sulfur oxidizing autotrophs as well as potential co-occurrence of Fe^III^ reducing autotrophs and S^0^ reducing heterotrophs. Putative S^0^ oxidizing populations in the amplicon data include *Thermocrinis* sp. (16.3% of Bacterial sequences recovered), *Thiomonas* sp. (14.0% of bacterial sequences recovered, [60]), and members of the Halothiobacillales (*Halothiobacillus* sp., *Thiovirga* sp., and uncultured sp., 17.5% of bacterial sequences recovered). These potential S^0^-oxidizing organisms may be associated with generation of EPS/biomass found encapsulating the spheres of S^0^ in the process of being oxidized (and consumed) (Figure 7C,D). There was recovery of bacterial sequences associated with putative Fe^III^ reducers Hydrogenothermales (*Sulfurihydrogenibium* sp. (6.1% of Bacterial sequences recovered), [61,62,63]) and archaeal sequences recovered associated with OTUs known to reduce S^0^ to sulfide in Desulfurococcales (*Acidilobus* sp. (21.9% of Archaeal sequences recovered), [64]) and Thermoproteales (Pyrobaculum sp., *Thermofilum* sp., and *Thermoproteus* sp. (74.5% of archaeal sequences recovered), [50,54,55]). This co-occurrence suggests these could represent the primary members of a putative consortium that could be driving observed Fe-S mineral precipitation (Figure 7E,F). These results suggest a complex sulfur cycle occurring within this hot spring where oxidation and reduction of S^0^ are co-occurring, similar to what we propose is happening in the Boulder Geyser outflow. Molecular results for Dante’s Inferno were consistent with previous work done at this site [65].

### 4.5. Photoautotrophy

Two sites below the upper temperature limit for photosynthesis (< 72 °C) with visible photosynthetic pigments were selected for sampling: ‘Avocado Spring’ and ‘The Dryer’ (Figure 1, Appendix A). At ‘Avocado Spring’, the grey and red mats (Figure 1) were sampled for carbon assimilation and microbial/geochemical characterization (in 2016 and 2018, respectively).

Light-dependent inorganic carbon assimilation was observed in red mats at ‘Avocado Spring’ in 2016 and 2018. Rates of assimilation that were higher in 2018 exhibited higher light dependent carbon uptake compared to 2016 (Figure 2 and Figure 3). Specifically, the red mat site sampled in 2018 exhibited 100% light dependent autotrophy (22.6 µg C uptake/g C biomass/hr), while the grey mat site sampled in 2016 exhibited light dependent autotrophy that was nearly half that of the red mat (12.9 µg C uptake/g C biomass/hr) and chemoautotrophic carbon uptake was also observed (3.3 µg C uptake/g C biomass/hr) (Figure 2, Appendix A). Consistent with photoautotrophy, 18.3% of bacterial sequences recovered from the red mat were associated with oxygenic phototrophs within the Cyanobacteria (e.g., Eurycoccales, Leptolyngbyales, Nostocales, Phormidesmiales, Psuedoanabaenales). No sequences were recovered from the grey mat that were affiliated with oxygenic phototrophs. Fe-S minerals were observed at both sites, suggesting a lack of oxygen (Figure 8). Approximately 6.9% of bacterial sequences recovered from the red mat were associated with potentially aerobic anoxygenic photoheterotrophs (Chloroflexales and Sphingomonadales), while a large number of archaeal sequences were affiliated with anaerobic heterotrophs/fermenters (e.g., Desulfurococcales (25.2%), Thermoproteales (12.8%), and Woesearchaeia (9.6%)), suggesting anoxic conditions within the mat. Most of the sequences of oxyphototrophs/cyanobacteria were uncultured, unclassified, or had low identities, suggesting the functions of the putative oxygenic phototrophs is poorly constrained. While we did not measure anoxygenic photosynthesis, this could be a primary mode of light-dependent primary productivity in ‘Avocado Spring’.

Recovery of both bacterial and archaeal sequences associated with nitrogen cyclers suggest a robust nitrogen cycle within the mat at ‘Avocado Spring’. Bacterial sequences (12.5%) included putative nitrate reducers and denitrifiers within the Betaproteobacteria, Caulobacterales, Chitinophagales, Cytophagales, and OLB14 as well as ammonia oxidizers in the Betaproteobacteria [66,67,68,69,70,71,72,73,74,75,76]. Archaeal sequences were predominantly Nitrosphaerales (34.2%), putative ammonia oxidizers [77,78,79]. Nitrogen cycling and loss through denitrification would explain the increase in δ^15^N values from the grey mat site closer to the source (1.35‰) to a more positive value in the down-stream red mat (4.08‰). 

Recovery of both bacterial and archaeal sequences associated with nitrogen cyclers suggest a robust nitrogen cycle within the mat at ‘Avocado Spring’. Bacterial sequences (12.5%) included putative nitrate reducers and denitrifiers within the Betaproteobacteria, Caulobacterales, Chitinophagales, Cytophagales, and OLB14 as well as ammonia oxidizers in the Betaproteobacteria [66,67,68,69,70,71,72,73,74,75,76]. Archaeal sequences were predominantly Nitrosphaerales (34.2%), putative ammonia oxidizers [77,78,79]. Nitrogen cycling and loss through denitrification would explain the increase in δ^15^N values from the grey mat site closer to the source (1.35‰) to a more positive value in the down-stream red mat (4.08‰). 

The black mat sampled in 2016 is enigmatic—most abundant bacterial and archaeal OTUs recovered could not be classified above the Order level. For instance, the 16S rRNA sequences assigned to bacteria contained 16.4% uncultured bacteria and 8.8% uncultured Chloroflexi SBR1031 (8.8%); while the 16S rRNA sequences assigned to archaea included uncultured Archaea (5.5% of archaeal sequences recovered), uncultured Bathyarchaeia (40.4%), unclassified Crenarchaeota (18.9%), uncultured Hadesarchaeaeota (8.5%), and unclassified Thaumarchaeota (12.3%), and Aigarchaeales (4.7%). Despite the observation of photoassimilation, no abundant photoautotrophs were observed in the amplicon library. Chemotrophic carbon uptake could be attributed to the bacterial sequences affiliated with the Acetothermiia (putative chemolithoautotrophs, [80]) and Thermodesulfobacteriales (chemoautotroph, [53]), and possibly archaeal sequences affiliated with Aigachaeales *Caldiachaeum* sp. (possible autotroph, [81]). The presence of putative sulfate reducers (including OTUs affiliated with Syntrophobacteriales *Thermodesulforhabdus* sp. and uncultured Deltaproteobacteria Desulfarculales) and Fe^III^ reducers (Thermodesulfobacteriales Caldimicrobium sp.) could explain the production of Fe-S minerals within the biofilm.

‘Avocado Spring’ grey and red mats had carbon uptake rates similar to those measured at the circum-neutral to alkaline chemotrophic sites Dante’s Inferno, Boulder Geyser, and ‘Rose Terrace Pool’ (Figure 2). This suggests the light-dependent carbon uptake rates at ‘Avocado Spring’ are substrate limited, as is assumed for the chemotrophic sites.

The green-pigmented sediments (Figure 1 and Figure 9) covering the old silica deposits associated with a transient phototrophic mat (Figure 10, Appendix A) at ‘The Dryer’ produced the highest carbon uptake rate of all sites (355 µg C uptake/g C biomass/hr), over an order of magnitude higher than the other sites (Appendix A). These high carbon uptake rates are assumed to be associated with oxygenic photosynthesis, which would not be substrate limited like chemotrophic and anoxygenic photoautotrophy. Very few bacterial sequences were recovered of known oxygenic phototrophs (Eurycoccales, 1.3%, Nostocales, 0.6%) or anoxygenic phototrophs (Chloroflexales, 1.0%). Photoassimilation may be due to the activity of phototrophic Eukarya. While we did not observe diatom frustules in the SEM imaging, we did recover chloroplasts sequences consistent with the presence of algae (data not shown). Most bacterial sequences recovered were associated with organotrophs (Acetobaterales, 9.6%, Chitinophagales, 56.1%, Thermales, 9.6%) or unclassified Bacteria (3.0%), while most archaeal sequences (90%) were associated with uncultured Sulfolobales (putative lithoautotrophs oxidizing sulfide, or heterotrophs, [46]). 

Presumably, the microbial community present during the deposition associated with relict mat textures had a significantly different composition than was present at the time of sampling (Figure 9, 10, Appendix A). Given the relatively short amount of time the thick phototrophic mat was present (~2 years, Appendix A) compared to the relatively large amount of silicified mat structure preserved, it is important to note the implications for preservation bias in these rapidly changing systems. Fewer OTUs affiliated with phototrophs and Archaea were recovered from the ‘The Dryer’ compared to the grey or red mats at ‘Avocado Spring’. This suggests that in these low biomass sites, while oxygenic photosynthesis may support higher carbon uptake rates per unit biomass, putative anoxygenic photosynthetic communities may support more diverse microbial populations.

C uptake in low biomass hydrothermal systems—a baseline of carbon fixation rates and implications for metabolisms in early Earth hydrothermal systems? Across a wide range of pH values (3.0 to 8.6) in high-temperature (> 72 °C) hot springs, chemolithoautotrophic C-uptake rates were similar, falling between 4.5 and 22.6 µg C uptake/g C_biomass_/hr, with an average value of 15.0 and a median of 17.4 for these systems that are inherently substrate limited. The consistent energy-deriving metabolisms across all of these sights are rooted in sulfur redox reactions, including oxidation of sulfur and sulfur compounds (e.g., sulfide, polysulfide, S^0^, S_2_O_3_^2−^) and reduction of sulfur and sulfur compounds (e.g., S^0^, S_2_O_3_^2−^, SO_4_^2−^). Furthermore, sulfur compound reduction to sulfide and metabolisms associated with the reduction of Fe^III^ to Fe^2+^ appear connected with production of Fe-S minerals (e.g., ‘Heartbeat Pool’, Dante’s Inferno, Boulder Geyser outflow). Redox reactions involving S^0^ seem particularly important. The prevalence of S^0^ in modern hydrothermal systems is linked to the oxidation of sulfide, which can occur abiotically and result in the production of S^0^ (e.g., [82]). However, the abiotic oxidation of sulfide is driven by the presence of free O_2_ in the atmosphere. Similarly, the abiotic oxidation of Fe^2+^ to Fe^III^ is driven by the presence of free O_2_. While microbially mediated processes driving Fe^2+^ and sulfide oxidation are known to occur, they are linked to either O_2_ (or for some NO_3_^−^) as the electron acceptor. Thus, the presence of Fe^III^ and S^0^ as potential energy sources for modern hot spring microbial communities cannot be completely decoupled from the presence of free O_2_ in the atmosphere. Furthermore, there is an abundance of allochthonous organic carbon sources produced via multicellular oxygenic photosynthesis-based ecosystems (e.g., the meadows and forests surrounding the Yellowstone hydrothermal areas). These factors raise potential doubt as to the chemical energy available to hot spring microbial communities: Is oxygen required to drive microbial chemolithoautotrophic metabolisms? Are heterotroph-dominant hot spring microbial communities supported by allochthonous input? 

Hot springs present on land surfaces during the early Archean (e.g., [1,2]) would have likely had sulfide/polysulfide and SO_3_^2−^ delivered via volcanic gases (e.g., [13]), but organic carbon would have likely been sourced solely from autochthonous sources. The presence of microbial communities capable of reduction and/or disproportionation of SO_3_^2−^, potentially linked to the oxidation of sulfide, would have provided a baseline for sulfur cycling early in Earth’s history, generating elemental sulfur and other mixed-valence sulfur compounds that could have served as further metabolic fuel for other members of the microbial community. Furthermore, production of sulfate through disproportionation of SO_3_^2−^ and atmospheric reactions of ultraviolet radiation with SO_2_ would produce SO_4_^2−^, which in the presence of sulfide can produce S_2_O_3_^2−^, another intermediate redox sulfur compound that can feed microbial metabolisms. A robust sulfur-cycling microbial community with chemolithoautotrophic members may serve as a model for those present in early Archaean hot springs. The occurrence of light-dependent carbon uptake in sulfide-containing hot springs (e.g., ‘Avocado Spring’) would add a further potential source for S^0^ and more oxidized sulfur compounds as well as autochthonous organic carbon generation following the advent of anoxygenic photosynthesis (e.g., [13]).

If we take modern chemolithoautotrophic carbon uptake in putatively substrate limited systems as a baseline for expected productivity in ancient hot spring microbial communities prior to the evolution of light-dependent carbon fixation, then the average value would range from 4.5 to 22.6 µg C uptake/g C_biomass_/hr. Following the evolution of anoxygenic photosynthesis, if we are to use the carbon uptake rates measured at ‘Avocado Spring’ as a proxy, we would in fact not expect a significant change in the maximum rate of carbon uptake (22.6 vs. 21.3 µg C uptake/g C_biomass_/hr), but the potential for increased microbial community diversity may help drive evolution as new niche space is generated. In fact, compared to the microbial community in the presence of oxygenic photosynthesis at ‘The Dryer’, while carbon uptake rates were over an order of magnitude higher, the microbial community diversity was much less than observed at ‘Avocado Spring’.

## 5. Conclusions

We observe chemoautotrophy that is fueled largely by redox reactions involving sulfur compounds as well as the potential for preservation of biosignatures, even in systems with very low carbon from biomass. Our data underscore emerging hypotheses for terrestrial life early in the Archean; including the predominance of sulfur-based autotrophic metabolisms in hot springs, an increase in diversity coincident with the emergence of anoxygenic photosynthesis, and energy substrate limitation of primary productively until the evolution of water-splitting oxygenic photosynthesis.

## Figures and Tables

**Figure 1 life-09-00064-f001:**
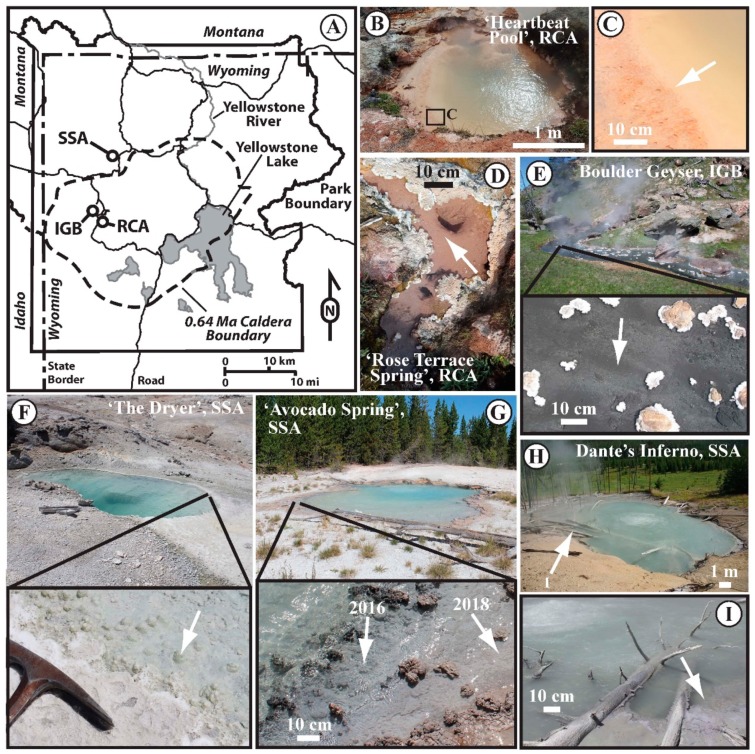
Map and images of sample site locations. (**A**) Map showing the hydrothermal areas sampled for this study: IGB = Imperial Geyser Basin of the Lower Geyser Basin, RCA = Rabbit Creek Area of the Midway Geyser Basin, SSA = Sylvan Spring Area of the Gibbon Geyser Basin. Sample site images clockwise from lower left: (**B**) ‘Heartbeat Pool’, RCA; (**C**) Closeup of sampling site at ‘Heartbeat Pool’, RCA; (**D**) ‘Rose Terrace Pool’, RCA; (**E**) Boulder Geyser with inset of outflow sampling area, IGB; (**F**) ‘The Dryer’ with inset of sampling area (rock hammer for scale), SSA; (**G**) ‘Avocado Spring’, SSA with inset of sampling area: grey mat sampled in 2016, red mat sampled in 2018; (**H**) Dante’s Inferno, SSA; (**I**) Closeup of sampling site at Dante’s Inferno. Arrows indicate location of sample collection. Scale bars provided in images without visual scale references present. Flow directions for samples collected where flow was occurring: ‘Rose Terrace Pool’—flow is from bottom to upper left; Boulder Geyser outflow (inset)—flow is from left to right; ‘Avocado Spring’—flow is from right to left, and for inset—flow is from left to right.

**Figure 2 life-09-00064-f002:**
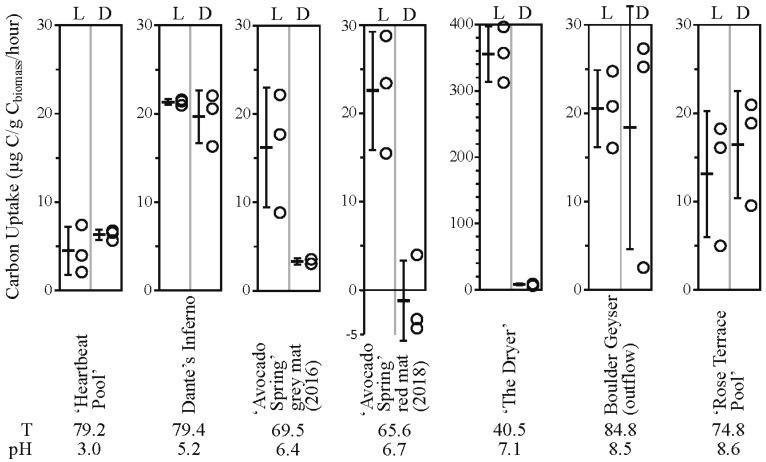
Carbon uptake experiment results in order (left to right) of increasing pH. L = light dependent (incubated in full light), D = dark treatment (incubated in the dark). Open circles are results of triplicate experiments, bars represent average values with error bars showing standard deviation. Note the difference in y-axis scale for ‘The Dryer’.

**Figure 3 life-09-00064-f003:**
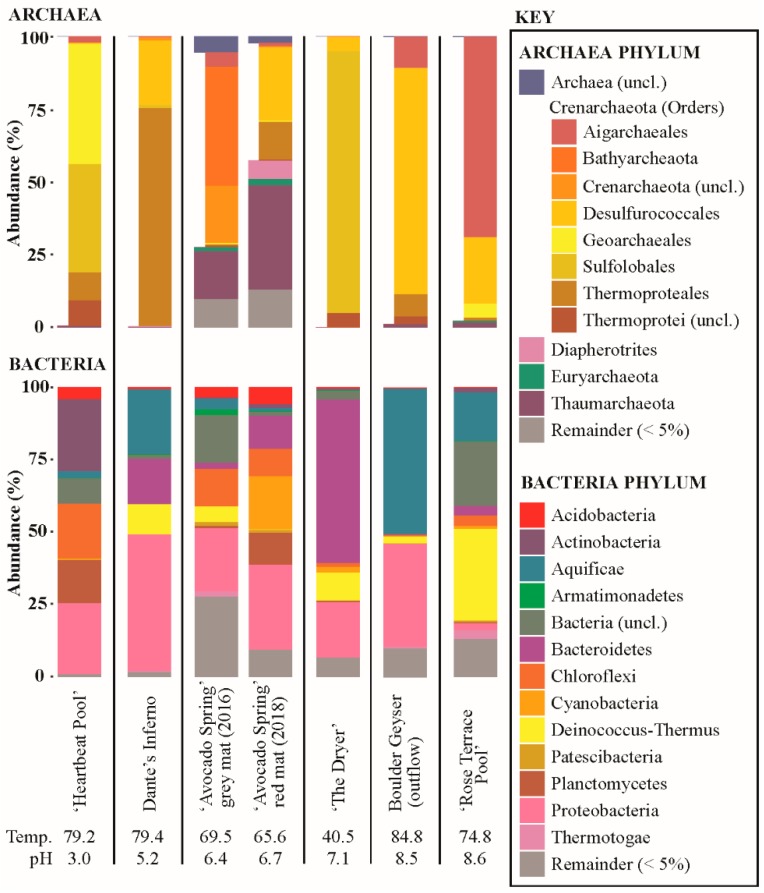
Molecular results of sites sampled and carbon uptake experiments conducted, in order (left to right) of increasing pH. Crenarchaeal Orders are presented in the space for the Phylum in the Archaeal sequences bar charts. Remainder = all sequences that account for less than 5% of recovered sequences for all sites. Uncl. = unclassified.

**Figure 4 life-09-00064-f004:**
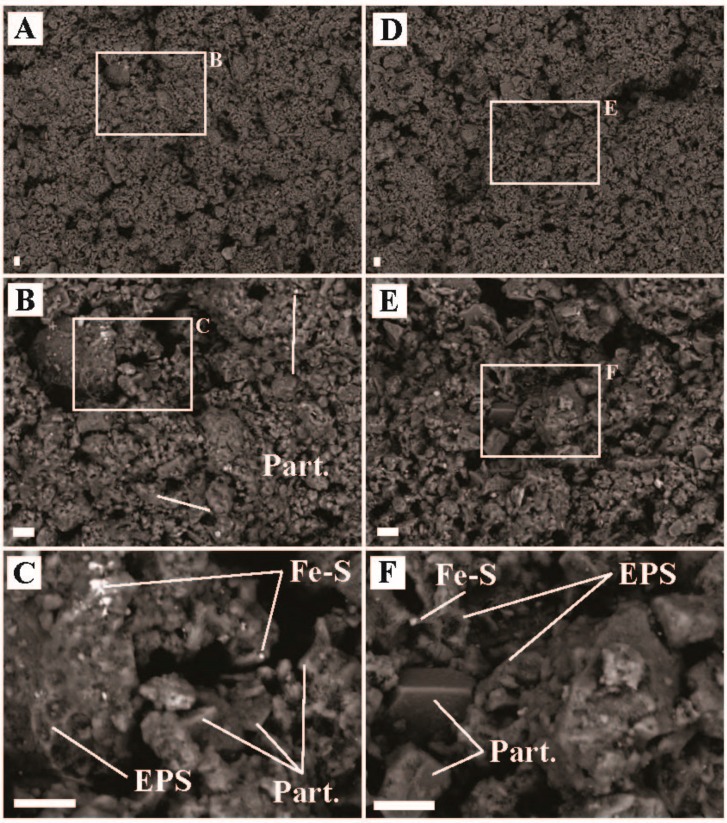
Scanning electron microscope images of sediment samples from ‘Heartbeat Pool’, Midway Geyser Basin, YNP (temperature = 79.2 °C, pH = 3.0). (**A**) Representative image of sediment textures, (**B**) inset of ‘A’ highlighting a region with biofilm and Fe-S minerals, (**C**) zoom in from ‘B’ highlighting physical relationships of mineral particulates (Part.), extracellular polymeric substance (EPS), and iron-sulfide minerals (Fe-S), (**D**) a separate region of representative sediment textures, (**E**) a zoom in from ‘D’ highlighting biofilm-mineral physical relationships, and (**F**) zoom in of ‘E’ showing Part., Fe-S, and EPS associations. All scale bars are 10 µm.

**Figure 5 life-09-00064-f005:**
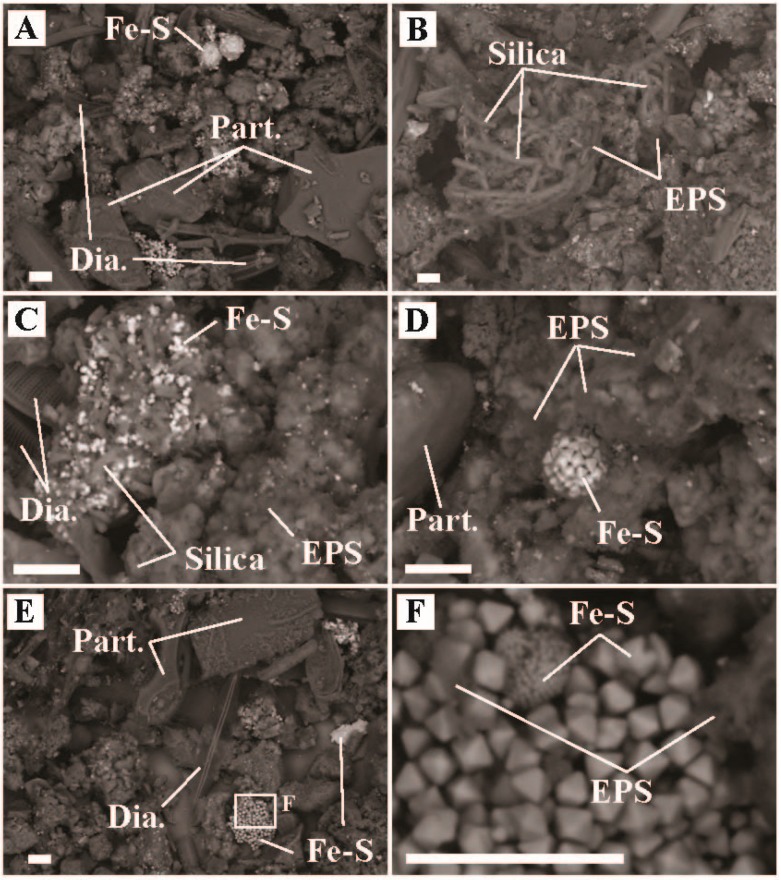
Scanning electron microscope images of sediment samples from Boulder Geyser outflow site, Lower Geyser Basin, YNP (temperature = 84.8 °C, pH = 8.5). (**A**) Representative image of sediment showing general physical relationships of particulates (Part.), allochthonous diatom frustules (Dia.), and iron-sulfide minerals (Fe-S), (**B**) image showing silica sheaths that have formed around microbial filaments and associated extracellular polymeric substance (EPS), (**C**) image highlighting physical relationships of EPS, iron-sulfide minerals (Fe-S), Dia., and silica precipitate (Silica), (**D**) image highlighting physical relationships between EPS, framboidal Fe-S, and Part., (**E**) image highlighting colonized Part., Dia., and framboidal and massive Fe-S, and (**F**) zoom in showing relationship of framboidal Fe-S and EPS from ‘E’. All scale bars are 10 µm.

**Figure 6 life-09-00064-f006:**
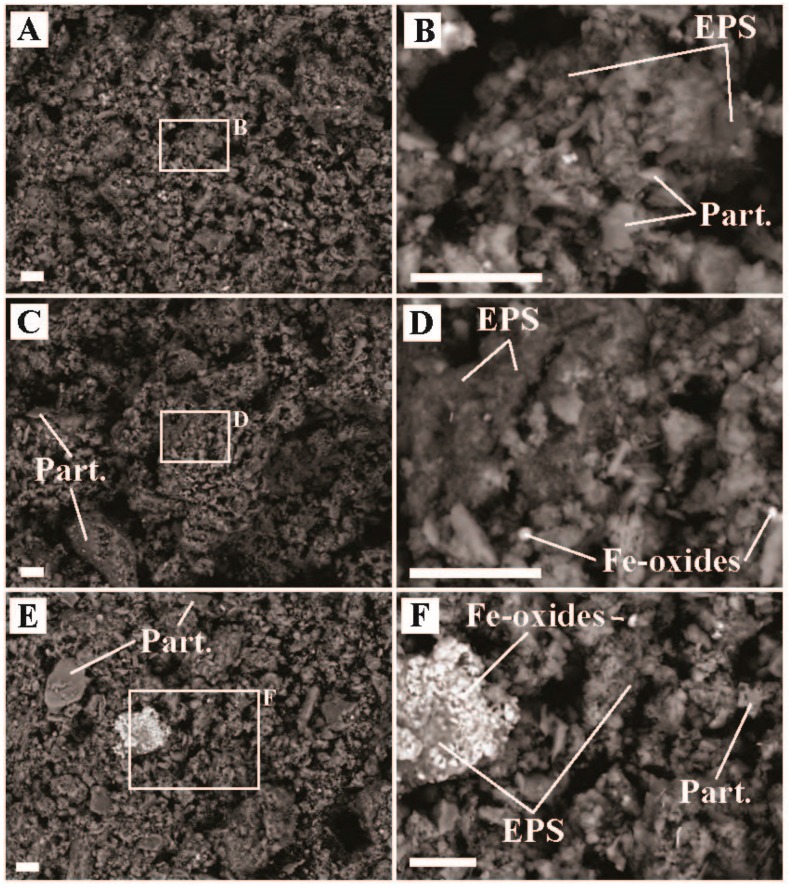
Scanning electron microscope images of sediment samples from ‘Rose Terrace Pool’, Midway Geyser Basin, YNP (temperature = 74.8 °C, pH = 8.6). (**A**) Representative image of sediment textures, (**B**) inset of ‘A’ highlighting a region showing physical relationship of particulates (Part.) and extracellular polymeric substance (EPS) in ‘A’, (**C**) representative sediment texture with larger Part., (**D**) inset of ‘C’ highlighting physical relationship of EPS and iron oxide minerals (Fe-oxide), (**E**) representative sediment texture showing physical relationship of larger particulates and a larger Fe-oxide mineral, and (**F**) inset of ‘E’ highlighting physical relationship of EPS, Fe-oxides, and Part. All scale bars are 10 µm.

**Figure 7 life-09-00064-f007:**
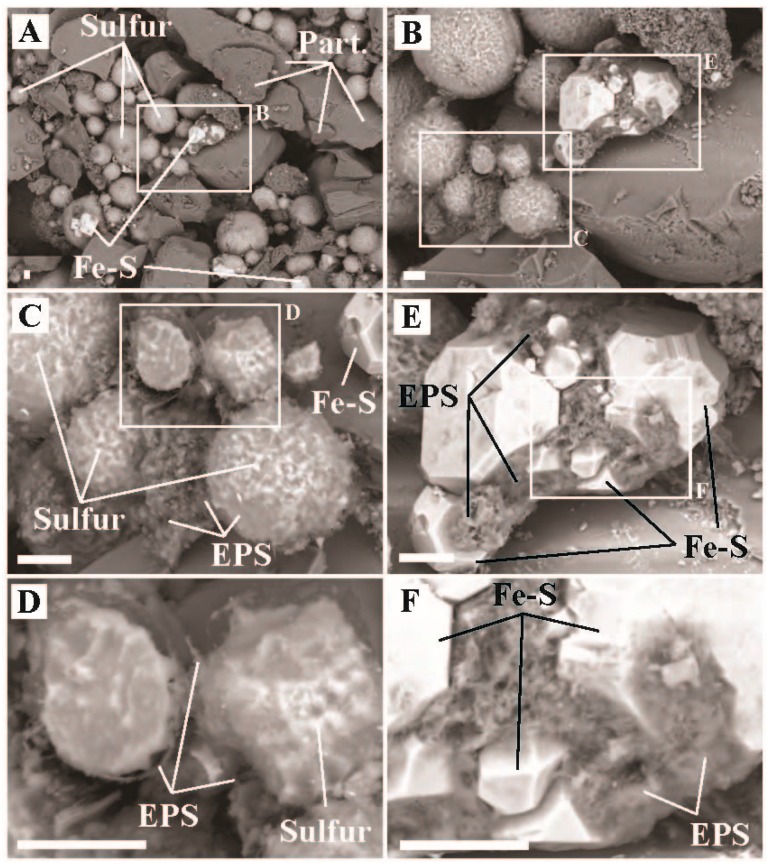
Scanning electron microscope images of sediment samples from Dante’s Inferno, Sylvan Spring Area, Gibbon Geyser Basin, YNP (temperature = 79.4 °C, pH = 5.2). (**A**) Representative image of sediment textures showing physical relationships of particulates (Part.), elemental sulfur (Sulfur), and iron-sulfides (Fe-S), (**B**) inset of ‘A’ highlighting a region with Sulfur and Fe-S, (**C**) zoom in from ‘B’ highlighting physical relationships of Sulfur and EPS, (**D**) inset from ‘C’ showing the association of EPS surrounding Sulfur and highlighting dissolution/oxidation of Sulfur, (**E**) inset from ‘B’ highlighting the physical relationship of EPS and Fe-S, and (**F**) zoom in of ‘E’ showing Fe-S and EPS associations. All scale bars are 10 µm.

**Figure 8 life-09-00064-f008:**
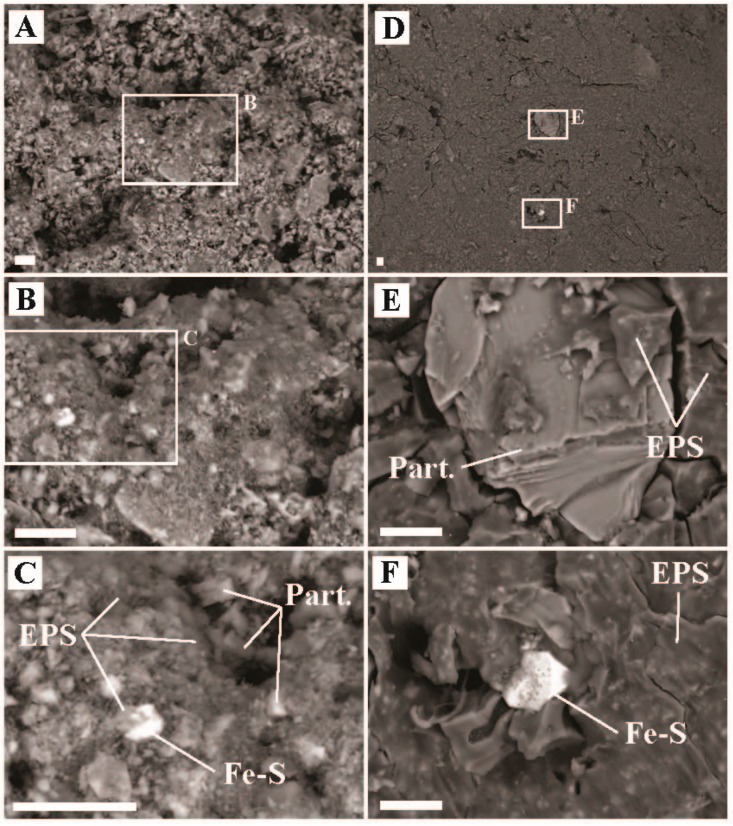
Scanning electron microscope images of sediment samples from ‘Avocado Spring’, Sylvan Spring Area, Gibbon Geyser Basin, YNP for left: 2016 (temperature = 69.5 °C, pH = 6.4), and right: 2018 (temperature = 65.6 °C, pH = 6.7). (**A**) Representative image of mat texture from the 69.5 °C site, (**B**) inset of ‘A’ highlighting a region with biofilm, (**C**) zoom in of ‘B’ highlighting physical relationships of mineral particulates (Part.), extracellular polymeric substance (EPS), and iron-sulfide minerals (Fe-S), (**D**) representative mat texture from the 65.6 °C site, (**E**) inset from ‘D’ highlighting the physical relationship of Part. and EPS, and (**F**) inset from ‘D’ highlighting the physical relationship of Fe-S and EPS. All scale bars are 10 µm.

**Figure 9 life-09-00064-f009:**
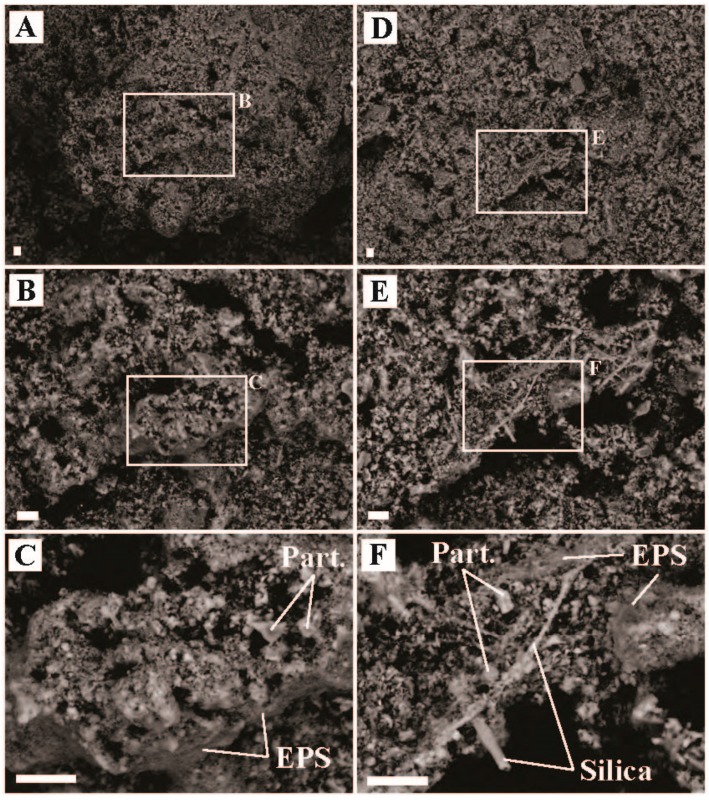
Scanning electron microscope images of sediment samples with an active phototrophic microbial community from ‘The Dryer’, Sylvan Spring Area, Gibbon Geyser Basin, YNP (temperature = 40.5 °C, pH = 7.1). (**A**) Representative image of surface sediment texture, (**B**) inset of ‘A’ highlighting a region with biofilm, (**C**) zoom in of ‘B’ highlighting physical relationships of mineral particulates (Part.) and extracellular polymeric substance (EPS), (**D**) representative mat texture from a different part of the sample, (**E**) inset from ‘D’ highlighting a region with silica-encrusted filaments, and (**F**) inset from ‘E’ highlighting the physical relationship silica sheaths (Silica), Part., and EPS. All scale bars are 10 µm.

**Figure 10 life-09-00064-f010:**
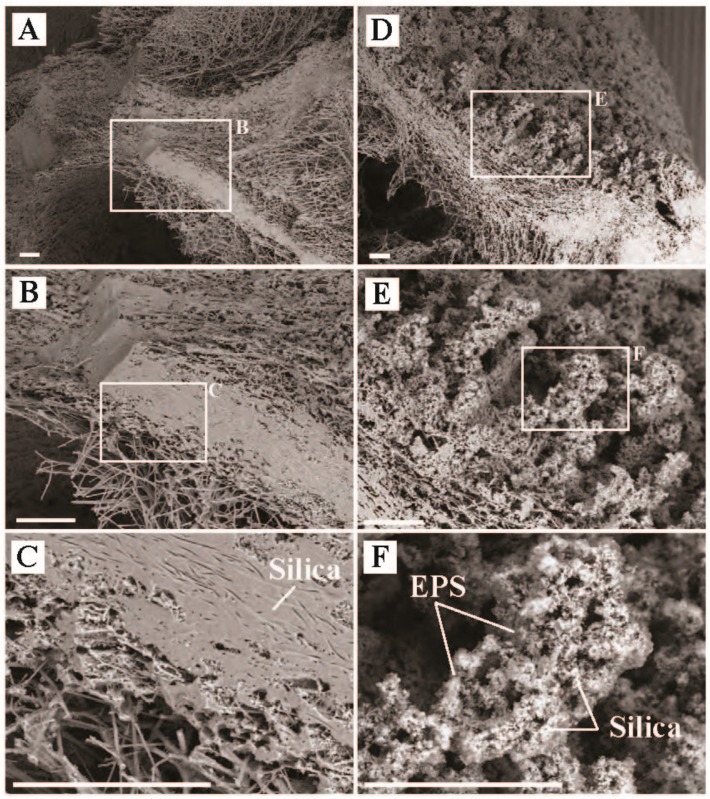
Scanning electron microscope images of silica precipitate samples from ‘The Dryer’ showing preserved photosynthetic mat textures, Sylvan Spring Area, Gibbon Geyser Basin, YNP. (**A**) Representative image of mat texture formed at ‘The Dryer’ during the 2010-2011 mat building period, (**B**) inset of ‘A’ highlighting the preserved mat and filament textures, (**C**) zoom in of ‘B’ showing preservation of areas where filaments resided during the active building of the mat textures, (**D**) representative mat texture from the surface showing the relationship of the active surface with the former mat underneath, (**E**) inset from ‘D’ highlighting the active surface texture, and (**F**) inset from ‘E’ highlighting the physical relationship of EPS and Silica. All scale bars are 10 µm.

**Table 1 life-09-00064-t001:** Physico-chemical properties of sampled hot springs and sediments/biofilms.

Sample Site	‘Heartbeat Pool’	Dante’s Inferno	‘Avocado Spring’		‘The Dryer’	Boulder Geyser OF	‘Rose Terrace Pool’
Hydrothermal Area	RCA, MGB	SSA, GGB	SSA	GGB	SSA, GGB	IGB, LGB	RCA, MGB
Date Sampled	06/06/18	07/24/16	07/24/16	06/05/18	07/24/16	06/29/17	06/06/18
YNP Inventory ID^a^	(none)	Dante’s Inferno	GSSGNN025		GSSGNN037	LRNN781	(none)
UTM	4929806	4949526	4949715		4949604	4933949	4929847
12 T	0515019	0518360	0518576		0518442	0512398	0515040
T (°C)	79.2	79.4	69.5	65.6	40.5	84.9	74.8
pH	3.01	5.19	6.39	6.71	7.13	8.47	8.62
Conductivity (µS/cm)	505	2250	1925	3596	2310	3280	2799
SiO_2(aq)_ (mmol/L)	4.34	2.91	2.55	4.32	2.19	2.86	4.83
Cl^−^ (mmol/L)	0.24 ± 0.00	15.94	6.57	7.26 ± 0.04	15.95	7.90 ± 0.04	6.30 ± 0.01
SO_4_^2−^ (mmol/L)	0.66 ± 0.00	2.13	3.27	1.71 ± 0.00	2.15	0.17 ± 0.01	0.14 ± 0.00
DIC (mmol/L)	*0.72 ± 0.06*	1.42	5.27	10.1	0.18	2.41	39.11 ± 3.26
DOC (µmol/L)	nd	195 ± 0.2	118 ± 0.4	nd	97.9 ± 0.2	31.9 ± 9.9	nd
Na^+^ (mmol/L)	0.81 ± 0.01	17.59	16.66	16.36 ± 0.34	17.61	14.89 ± 3.98	11.06 ± 0.17
K^+^ (mmol/L)	0.17 ± 0.01	1.53	0.68	0.46 ± 0.03	0.95	0.30 ± 0.08	0.23 ± 0.01
Mg^2+^ (µmol/L)	7.90 ± 0.17	0.83	0.68	0.52 ± 0.32	3.57	0.31 ± 0.08	0.21 ± 0.17
Ca^2+^ (µmol/L)	37.27 ± 1.14	106.16	80.64	96.93 ± 1.31	128.98	28.55 ± 7.36	14.31 ± 0.72
Sulfide (µmol/L)	7.8	538.0	313.0	46.0	78.0	45.5	1.0
Fe^2+^ (µmol/L)	bdl	0.25	0.10	bdl	bdl	0.90	bdl
P (µmol/L)	2.40 ± 0.10	nd	nd	8.96 ± 0.05	nd	4.25 ± 0.08	6.57 ± 0.05
Al (µmol/L)	42.01 ± 0.08	12.59	5.56	4.42 ± 0.05	3.38	12.06 ± 0.14	11.64 ± 0.03
Mn (nmol/L)	675 ± 0.8	684	218	206 ± 1	475	30.1 ± 0.08	7.71 ± 0.14
Fe(T) (nmol/L)	507 ± 5	169	85	252 ± 16	381	384 ± 7	111 ± 2
As (nmol/L)	137 ± 16	33915	14338	nd	34269	15229 ± 170	16951 ± 94
Biomass C (%)	0.672	0.081	0.586	0.451	0.269	1.099	0.118
Biomass N (%)	0.049	0.016	0.060	0.061	0.034	0.056	0.010

Names in single quotes denote colloquial names used solely for distinguishing sites, and are not official or recognized by the National Park Service. OF = outflow, SSA = Sylvan Spring Area, GGB = Gibbon Geyser Basin, RCA = Rabbit Creek Area, MGB = Midway Geyser Basin, IGB = Imperial Geyser Basin, LGB = Lower Geyser Basin, DIC = dissolved inorganic carbon, DOC = dissolved organic carbon, nd = not determined, bdl = below detection limits. Analytical methods used are described in the methods section of the text. Stable carbon isotope values for DIC, DOC, and stable carbon and nitrogen isotope values for biomass are provided in Table 2. Standard deviations are given when available. aSite designations listed on the Yellowstone National Park Research Coordination Network website (http://www.rcn.montana.edu/Default.aspx). Values in italics are semi-quantitative (low signal).

**Table 2 life-09-00064-t002:** Stable carbon and nitrogen isotope values of sampled hot springs and sediments/biofilms.

Sample Site	‘Heartbeat Pool’	Dante’s Inferno	‘Avocado Spring’		‘The Dryer’	Boulder Geyser OF	‘Rose Terrace Pool’
Sample Description	Sediments	Sediments	Grey Mat (‘16)	Red Mat (‘18)	Silica Precipitate	Sediments	Sediments
T (°C)	79.2	79.4	69.5	65.6	40.5	84.9	74.8
pH	3.01	5.19	6.39	6.71	7.13	8.47	8.62
DIC δ^13^C (‰)	*−8.44 ± 0.08*	−0.51 ± 0.12	0.74 ± 0.12	−0.46 ± 0.08	3.06 ± 0.12	−1.17 ± 0.08	−1.65 ± 0.08
DOC δ^13^C (‰)	nd	−22.16 ± 1.00	−22.55 ± 0.05	nd	−27.80 ± 0.09	−24.47 ± 0.89	nd
Biomass δ^13^C (‰)	−25.59 ± 0.06	−14.69 ± 1.58	−22.87 ± 0.73	−15.83 ± 0.06	−16.67 ± 0.96	−24.33 ± 0.06	−24.33 ± 0.06
Biomass δ^15^N (‰)	−2.26 ± 0.12	0.24 ± 0.12	1.35 ± 0.12	4.08 ± 0.12	−4.13 ± 0.12	1.93 ± 0.12	2.50 ± 0.12

DIC = dissolved inorganic carbon, DOC = dissolved organic carbon, nd = not determined. Values in italics are semi-quantitative (low signal). δ^13^C values are versus Vienna Pee Dee Belemnite (VPDB), and δ^15^N values are given versus atmospheric air.

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
