# Peer review of "Productivity and Community Composition of Low Biomass/High Silica Precipitation Hot Springs: A Possible Window to Earth’s Early Biosphere?"

_life, 2019, doi:10.3390/life9030064_

Reviewer 1 Report

The manuscript “Productivity and community composition of low biomass/high silica precipitation hot springs: a possible window to Earth’s early biosphere?” provides insight into the biomass and metabolisms present in selected springs in the Yellowstone system. Overall, the data are reasonable and worthy of publication. However, the writing is not well-organized, and the arguments are not well-supported. The following are the major comments. Minor comments are on the manuscript.

Major comments:

1.      The manuscript needs better organization. For example, the site descriptions include results and interpretations. Likewise, results include interpretations, and the bulk of the discussion should be in an interpretation section. A better way to descibbe the sites would be to include field methods in the Methods section and then describe the sites in a consistent format.

2.      One of the major points that is consistently made is that the oxidation of sulfur precludes the formation of iron sulfide minerals. This is not true. Pyrite can precipitate under fairly high eH conditions, and going from pyrrhotite to pyrite involves an oxidation of S. In order to make this statement, the specific mineral phases that are present would need to be shown. This leads into the next major comment.

3.      SEM images are out of focus, and not discussed in methods. What instrument did you use? What vacuum? Were the samples coated and with what? What KeV? What are the particulates? Are these detrital or authigenic minerals? What is the chemistry of the different features, i.e., how is the EPS determined? By a high C component or just because it looks like it might be EPS? In Figure 6, why are the bright minerals called iron oxides rather than pyrite? What is the evidence that the spheres are S? What is the evidence that the spheres are being oxidized? Use the EDX to determine what the elemental composition of the different features is. Quantify the observations of the minerals of interest, e.g., does “scattered” mean common, present, or rare? Or does it mean a self-organized spacing as opposed to a random spacing?

Reviewer 2 Report

I considered the research to be meticulous and well presented. My only comment is that early life most probably commenced around ocean floor hydrothermal vents for many reasons not least the paucity of land areas in the early Archean but this is not relevant to the purpose of the paper.

Author Response

Round  2

Reviewer 1 Report

The authors have sufficiently addressed the review and I recommend publication. 

Author Response

The Reviewer as accepted the revisions made, and has not recommended any new edit suggestions.

Reviewer 1: "The authors have sufficiently addressed the review and I recommend publication."

Again, we thank the reviewer for all of the great edits and suggestions that have greatly improved the quality and content of the manuscript.